# The Differential DNA Hypermethylation Patterns of microRNA-137 and microRNA-342 Locus in Early Colorectal Lesions and Tumours

**DOI:** 10.3390/biom9100519

**Published:** 2019-09-21

**Authors:** Elham Kashani, Mahrooyeh Hadizadeh, Vahid Chaleshi, Reza Mirfakhraie, Chris Young, Sanaz Savabkar, Shiva Irani, Hamid Asadzadeh Aghdaei, Maziar Ashrafian Bonab

**Affiliations:** 1Institue of Pathology, University of Bern, 3010 Bern, Switzerland; elham.kashani@pathology.unibe.ch; 2Basic and Molecular Epidemiology of Gastrointestinal Disorders Research Center, Research Institute for Gastroenterology and Liver Diseases, Shahid Beheshti University of Medical Sciences, Tehran 19839-63113, Iran; chaleshi@gmail.com (V.C.); sanaz.savabkar@gmail.com (S.S.); hamid.asadzadeh@sbmu.ac.ir (H.A.A.); 3Department of Applied Sciences, University of the West of England (UWE-Bristol), Bristol BS16 1QY, UK; mahrooyeh.hadizadeh@gmail.com; 4Department of Medical Genetics, Shaheed Beheshti University of Medical Sciences, Tehran 19839 69411, Iran; reza_mirfakhraie@yahoo.com; 5Leicester School of Allied Health Sciences, Faculty of Health and Life Sciences, De Montfort University, Leicester LE1 9BH, UK; chris.young@dmu.ac.uk; 6Department of Biology, School of Basic Sciences, Science and Research Branch, Islamic Azad University, Tehran 1477893855, Iran; shi_irani@yahoo.com

**Keywords:** colorectal cancer, microRNA, methylation, miRNA-137, miRNA-342, precursor colon/rectum polyps

## Abstract

Colorectal cancer (CRC) is the third most commonly diagnosed cancer worldwide, representing 13% of all cancers. The role of epigenetics in cancer diagnosis and prognosis is well established. MicroRNAs in particular influence numerous cancer associated processes including apoptosis, proliferation, differentiation, cell-cycle controls, migration/invasion and metabolism. MiRNAs-137 and 342 are exon- and intron-embedded, respectively, acting as tumour-suppressive microRNA via hypermethylation events. Levels of miRNAs 137 and 342 have been investigated here as potential prognostic markers for colorectal cancer patients. The methylation status of miRNA-137 and miRNA-342 was evaluated using methylation-specific (MSP) polymerase chain reaction (PCR) on freshly frozen tissue derived from 51 polyps, 8 tumours and 14 normal colon mucosa specimens. Methylation status of miRNA-137 and miRNA-342 was significantly higher in tumour lesions compared to normal adjacent mucosa. Surprisingly, the methylation frequency of miR-342 (76.3%) among colorectal cancer patients was significantly higher compared to miR-137 (18.6%). Furthermore, normal tissues, adjacent to the lesions (N-Cs), displayed no observable methylation for miRNA-137, whereas 27.2% of these N-Cs showed miRNA-342 hypermethylation. MiRNA-137 hypermethylation was significantly higher in male patients and miR-342 hypermethylation correlated with patient age. Methylation status of miRNA-137 and miRNA-342 has both diagnostic and prognostic value in CRC prediction and prevention.

## 1. Introduction

Colorectal cancer (CRC) is a major health burden comprising 13% of new cancer cases diagnosed worldwide annually, ranking as the third most prevalent and fourth most frequent cause of cancer-related mortality. CRC is responsible for more than 600,000 deaths each year and morbidity and mortality rates continue to increase, partly due to lack of appropriate and reliable early detection methods [1,2,3,4]. In the majority of cases, CRC occur sporadically and is not related to genetic predisposition or family history. However, 20–30% of patients have a positive family history of CRC and 5% of these patients show Mendelian inheritance [5].

Early detection of non-invasive tumors using DNA/RNA-based methodologies remains an exciting and emerging area of research into CRC screening methodologies. These methods are essential tools for researchers and practitioners to characterize the noninvasive tumors, plan for individualized treatments and track tumor treatment efficacy and prognosis in CRC. Of such technics, gene-specific DNA methylation patterning is perhaps the most attractive [6]. DNA methylation mainly occurs in CpG islands and as DNA methylation is more stable in comparison to mutations, it has been considered as a favorable area for biomarker exploration and identification [1,5,6].

Genes affected by aberrant hypermethylation are not confined to classical coding-based tumour suppressors; non-coding sequences such as microRNA genes are also affected and are significant players in tumorigenesis [7,8]. DNA methylation has received significant recent attention as a promising tool for cancer detection and prevention with promise for reducing mortality rates [8,9,10,11].

Eslamizadeh et al. (2018) [12] reported that microRNA expression levels correlate with different stages of CRC, hence they could provide new insight into tumour development, whilst being suitable and reliable biomarkers for CRC diagnosis and prognosis. Indeed, Feinberg and Vogelstein (1983) [13] previously showed DNA hypomethylation (mainly at CpG islands) at the early stages of CRC, combined by hypermethylation and inactivation of tumor suppressor or DNA repair genes.

Epigenetic changes have recently been associated with the normal mucosa aberrant crypt focus (ACF)-adenoma-carcinoma sequence, which itself plays an important role in CRC development [14]. DNA methylation status, therefore, appears to be a fundamental factor in carcinogenesis of CRC process. Specifically, it has previously been shown that hypermethylation of MMR genes results in MSI-sporadic CRC; through CpG Island Methylator Phenotype (CIMP), this was linked with chromosomal instability (CIN) in colon malignancy (where promoter methylation of GATA4, GATA5, p16 was previously shown to result in chromosomal loss or gain); changes in methylation status have been reported in a variety of genes relevant to these particular pathways [15,16,17,18,19,20]. Moreover, to date many studies have highlighted the potential benefits of implementing DNA-based aberrant methylation status assays in different biologic fluids as strategies for colorectal carcinoma early detection [21]. Lyberopoulou et al. (2017) [22] studied the methylation profile of CRC-specific genes, vimentin (VIM) and secreted frizzled-related protein 2 (SFRP2) in circulating tumor cells’ (CTCs) DNA. The authors confirmed essential roles for VIM and SFRP2 in important signaling pathways such as growth, proliferation, invasiveness, epithelial to mesynchymal (EMT) phenotype and stemness, using CTCs of CRC patients. They also showed CTCs to be a promising tool in cancer diagnostics, particularly for VIM (76.9% methylated, compared to the 53.8% in tissue samples).

It has been proposed that cancer cells rewire DNA methylation patterns of specific genes (either in their promoter region or in the gene body) to induce expression of oncogenes, to support their aggressiveness and silence the expression of tumor suppressors, for instance, to surmount apoptosis. This phenomenon has been specifically (but not exclusively) reported in context of CRC [22].

MicroRNAs (miRNAs = miRs) control >30% of regulatory pathways related to human gene expression and >500 miRs have been identified to date [23]. Growing bodies of evidence suggest that miRNAs are commonly dysregulated in human malignancies including CRC [11,22,23,24,25].

miRNA-137, a 23 bp, highly conserved miRNA among vertebrates [26,27] is one of the 10% intragenic miRNAs located in 1p21.3 and is embedded within an expanded CG island of a non-coding gene whose transcript is a lncRNA (AK094607; also called BRAMY2014205 and MIR137HG, counted as primary miR-137 coding sequence) [28,29]. Its downregulation due to proximal promoter hypermethylation has been reported in many cancer types including gastric cancer [30,31,32] melanoma [33] neuroblastoma [34,35] squamous cell carcinoma of head and neck (SCCHN) [36,37,38,39] and CRC [40,41,42,43]. MiRNA-342 (a 21bp miRNA) is encoded within an intron of a protein-coding gene (*EVL*) on chromosome 14q32.2. The 5`UTR of *EVL* contains a dense CpG island. It has been proposed previously that methylation status could be utilized as a non-invasive CRC biomarker [43,44,45,46,47,48]. MiR-137 and miR-342 are integral microRNAs implicated in as MSI-targeting microRNAs in CRC [49]. Taking into consideration that MSI status is one of the fundamental main initial events in carcinogenesis of colorectal lesions and tumor budding, we hypothesized the possibility of these two microRNAs deregulation may be useful as a biomarkers for the early detection of pre-cancerous and malignant lesions. As the role of miRNA-137/-342 promoter hypermethylation in colorectal lesions has not been fully investigated properly, the preliminary objective of this study was to evaluate the methylation status of miR-137 and miR-342 in different precursor and tumour lesions of colorectal tissues, tracing any possible correlations of this occurrence with CRC risk factors and assessing any significant patterns of methylation between CRC and normal colonic tissues.

## 2. Materials and Methods

### 2.1. Sampling

Specimens were obtained with patient consent during September 2016 to February 2017 from random patient samples referred to the Research Institute for Gastroenterology and Liver Diseases (RIGLD), Taleghani Hospital, Shahid Beheshti University of Medical Sciences, Tehran, Iran, who had undergone a colonoscopy procedure for either screening or polypectomy purposes. All procedures performed in this study involving human participants were in accordance with the ethical standards of the institutional research committee and with the 1964 Helsinki declaration and its later amendments or comparable ethical standards. Ethical approval was accepted on April 2016 by the ethical board of the Research Institute of RIGLD (project identification code: 2347-231) and Informed consent was obtained from all individual participants included in the study. In this study, 59 endoscopically-derived lesions and 14 normal colonic mucosas were evaluated. Sampling was done randomly during colonoscopy procedures, disregarding of lesion size, anatomic site and number of lesions in the affected colon. The specimens were flash frozen in liquid nitrogen and stored at −80 °C for further evaluation. Non-Persian patients and those who had positive history of chemo and radio therapies or cancer-related surgeries were dismissed according to study exclusion criteria.

### 2.2. Genomic DNA Extraction

DNA extraction was performed utilizing Qiagen QIAamp DNA Mini Kit (Cat. 51324, Qiagen, Germany) according to the manufacturer’s instructions. The concentration of the extracted DNA was measured by Nanodrop (Labtech; UK) @260 and 280 nm.

### 2.3. Sodium Bisulfite Treatment

Each DNA template was incubated with sodium bisulfite solution in order to achieve two distinctive and detectable template strands which are different merely upon their methylation status using Qiagen EpiTect Bisulfite Kit (Cat. 59104, Qiagen, Germany). 

### 2.4. Methylation-Specific Polymerase Chain Reaction (MSP)

Methylation-specific polymerase chain reaction (MSP) was conducted using two primer pairs; each specifically designed to attach to methylated or unmethylated strands. Segments were amplified in a total volume of 12.5 µL (Table 1) according to reactions profiles detailed in Table 2. Sequence, relative length, GC content and annealing temperature of primers are shown in Table 3 [27]. 

CpG-methylated HeLa Genomic DNA (New England BioLabs; Cat.#N4007S) was utilized as positive control for methylation, negative control was fully unmethylated DNA (EpiTect PCR control DNA; Qiagen). Two separate temperature gradients were performed to assess optimal annealing temperature for both forward and reverse primers (miR-137; 57 °C, miR-342; 63 °C). The specific band size observed was 140 bp and 210 bp, respectively. PCR products were run on 2% agarose gels and visualized via EtBr fluorescence (Figure 1).

### 2.5. Statistical Analysis

Dataset was analysed using SPSS.21 (manufacturer), utilizing descriptive analysis (including chi-square and Fisher`s exact test). Possible significant correlations between methylation status of miR-137/-342 and demographic, pathologic, clinical, lifestyle and symptom-related features were thereby evaluated (Appendix A). Selected features were chosen following consultation with an epidemiologist expert and were based on World Health Organization (WHO) criteria for CRC [50]. Furthermore, the presence of any significant correlations between methylation of miR-137 and miR-342 and quantitative traits such as patient’s age and BMI was evaluated using independent *t*-test (Appendix A). Normalization of the quantitative traits was assessed using one sample Kolmogorov–Smirnov test. Traits were also compared between distinct polyp sub-types using one-way analysis of variance (ANOVA). 

## 3. Results

The clinical, pathological, demographic and other characteristics of participants in this study are shown in Table 4. Despite randomized sampling, the majority of samples (39%) were 1–10 mm diameters and were the only lesions detected in the colon and rectum as a whole. Furthermore, the majority of the patients were Farsi ethnics, primarily educated and not addicted to opium, smoking or alcohol (Table 4 and Appendix A).

Synchronous assessment of dysplasia degree and the histological-pathological state of the samples showed that except for invasive adenocarcinomas (concordant with tumour lesions), the majority of the lesions had lower degrees of dysplasia, disregarding their defined sub-types (Figure 2), this finding suggests that most cases evaluated in this study were in early carcinogenesis. According to our findings, miR-137 and miR-342 methylation was significantly higher in tumour lesions in comparison with their normal adjacent mucosa *p* = 0.002 and *p* = 0.02, respectively. The methylation frequency among patients was 18.6% for miR-137 and 76.3% for miR-342. Although 27.27% of normal adjacent mucosa specimens were also displayed methylated miR-342 regions, none showed methylation of for miR-137.

Normal specimens derived from non-lesion normal controls were entirely unmethylated. Furthermore, we detected a significantly higher frequency of miR-137 methylation among males in comparison with female CRC patients (*p* = 0.002). Also, there was a statistically significant positive correlation between dysplasia degree and methylation status of miR-137 (0.016); 8.6% of the low grade-dysplastic samples depicted methylated state. In addition, patients with positive family history of colorectal cancer or other gastrointestinal-related cancers showed significantly higher frequencies of miR-137 methylation in comparison with patients with no family history (*p* < 0.0001).

For miR-342, independent *t*-test results indicated an age-related hypermethylation pattern (*p* < 0.001). miRNA-342 status was also in positive correlation with the presence of chronic constipation in patients (*p* = 0.021). The crosstabs evaluated are shown in Appendix A. Comparing the miR-137 methylation status between different sub-types of lesions (according to their pathological report sheet), we observed a higher methylation frequency among villous-type polyps (Table 5); an observation that proposed miR-137 methylation could at least in a part depend on distinct aetiology and pathway of villous-lesion formation (in comparison to non-villous types) and to a lesser degree, on minor on dysplasia state and degree. On the other hand, we observed high miR-342 methylation frequency for hyperplastic polyps (Table 5). These results strongly suggest distinct aberrant methylation status positively correlating with CRC progression in miRNAs 137 and 342.

Synchronous assessment of both miRNA loci together suggested 20.3% of patients were methylation-intact for both genes, 64.4% were methylated for at least one of these loci and 15.3% were methylated for both.

## 4. Discussion

In the recent era of research, efforts to configure and validate miRNA-based biomarkers have evolved significantly due at least in a part to the fact that in comparison with classic gene biomarkers, their alternation is much more tissue-specific. As a result, smaller evaluations can provide robust indications of promising detection and prognostic methodologies. Their application is also more accurate and less costly [44,51,52,53,54,55].

miRNA biomarker validation must involve thorough assessment using distinct precursor lesion types. Therefore, we have provided evidence for the exon-embedded miR-137 and intron-embedded miR-342 locus methylation status in tumour and polyp lesions of colon and rectum samples with distinct pathological sub-types. Furthermore, in a case-control study, we compared this status between tumour lesions and their adjacent normal mucosa. We also analysed normal colonic mucosa of a small population of lesion-free, unaffected patients. In addition, an accurate, comprehensive assessment of a possible correlation between demographic, pathological, clinical, lifestyle and symptom-related features was conducted. Tumour suppressive roles of miR-137 and miR-342 and their epigenetic roles in oral cancer and glioblastoma cell lines have been demonstrated previously [56,57,58,59,60]. Although there are very few previous studies to which the role of miR-137 and miR-342 in triggering CRC is referred, their downregulation has been reported in colon and gastric cancers [30,40]. Interestingly, upregulations of miR-137 and miR-342 have been reported for oral squamous cell carcinoma and acute promyelocytic leukaemia [61,62,63,64].

Chen et al. [65] through a follow-up cohort study demonstrated that miR-137 normal expression levels correlate with better survival rates in CRC patients. This has also been suggested for SCCHN patients and stage IV melanoma patients [36,66,67]. Ariel Bier et al. (2013), [59] detected promoter hypermethylation of miR-137 in glioblastoma specimens. Although their relative sample size was small (8 glioblastoma and 8 relevant normal), their methylation analysis technique was truly locus-specific and quantitative, with higher specificity in comparison with the MSP technique we utilized in the current study. However, this study used DNA extracted from formalin-fixed paraffin-embedded (FFPE) tissues and it is believed that this would negatively influence the actual methylation status of the tissue [67,68].

In another study by Balaguer et al. (2010) [40], average methylation levels were reported as 33.09% for adenomas, 31.67% for CRCs, 10.27% for normal adjacent mucosa (N-Cs) and 7.7% for normal mucosa controls (N-Ns), respectively. Concordant with our study, methylation frequency of miR-137 in tumours was significantly higher than their adjacent normal tissues (N-C) (*P* < 0.0001). This observation indicates cancer-specificity of miR-137 (i.e., Methylation of miR-137 preferably happens in neoplastic tissues and the methylation observed in normal mucosa is not due to aging) [69,70,71,72,73,74]. In both studies, Methylation of miR-137 was not concordant with aging for N-N or N-C specimens. Incidence of methylation in N-Cs and not in N-Ns could be an indication that methylation-related field defects may exist in CRC.

In our study, we did not detect any significant differences in methylation levels of polyp and tumour specimens (*p* = 0.141) which is concordant with the Blaguer group’s study (*p* = 0.8352); this observation recommends that methylation of this miR-137 is an early event in colorectal carcinogenesis. Taking into account that epigenetic modifications are highly influenced by lifestyle and geographical traits of different populations, we excluded patients of non-Persian nationality from our study. Which is considered an improvement on the Balaguer et al. study [40], which utilized samples taken from two distinct populations (Japanese and Spanish), a factor that certainly causes errors and bias in referring the methylation status solely to cancer (especially due to indistinct mixture proportions of two populations).

Tissue-specificity of miRNA-137 has been previously and separately demonstrated [29,40], in colorectal cancer and oral lichen planus (OLP) syndrome, confirming that miRNA-137 could have value for CRC screening. MiR-137 was selected for this study due to its fundamental role in cell cycle control [71]; in addition, promoter hypermethylation of miR-137 in colorectal carcinoma tissues had been reported [40,41]. Langevin et al. in 2010 showed significant correlation between miR-137 methylation status and female sex (*p* = 0.007) [38,39]. Inversely, in the current study this event was shown to be correlated with male sex (*p* = 0.002), which is in agreement with the Dang et al. study [29]. This illustrates cancer-specificity of miR-137 epigenetic signatures despite its common down-regulation in most of the cancer types. Therefore, the level of this down-regulation and its correlation with demographic factors such as sex and age has different implications in different cancer types. Gender-related hypermethylation could be due to sex hormone effects and physiological differences among target organs in both sexes.

To our knowledge, no study has previously shown any correlation between family history of CRC and degree of dysplasia with methylation status of miR-137. Bandres et al. (2009) [72] evaluated the methylation status of miR-137 in colorectal cancer cell lines and primary colorectal tumours and their adjacent normal tissues using MSP technic [72,73,74,75] identifying field defects in colorectal carcinogenesis. Regarding the high frequency of hypermethylation in miRNA loci, they proposed aberrant microRNA-methylation as a promising and putative tumour marker. MiR-137 methylation was observed for all tumour specimens (31 of 31) and 23% of (7 of 31) normal adjacent tissues. However, we observed 37.5% of tumour specimens to be methylated and no methylated normal adjacent specimens were observed. Unfortunately, this study did not manage to evaluate polyps which are defined as precursors of CRC.

Grady et al. (2008) [75] reported that has-miR-342 expression was silenced due to aberrant methylation in a CpG island of the 5` region of its target sequence (EVL). They conducted MSP on CRC and non-CRC cell lines separately and methylation of miR-342 was exclusively seen in CRC cell lines [75,76]. They then used MSP to assess the miR-342 methylation status of 9 adenomas, 42 CRC samples and their normal adjacent mucosa. They observed that 76% of tumours and 67% of adenomas were methylated; an observation that was in concordance with our results. They also detected 56% methylation in their normal adjacent mucosa, an observation that indicates methylation-related field defect phenomenon and is ascertained by our study. No demographic features were assessed in their study.

Age-related hypermethylation and miRNA-dysregulation due to aging have been reported for many genes in various cellular contexts [77,78,79,80,81,82,83,84,85]. Moreover, emerging evidence supports the existence of epigenetic changes in other age-related conditions such as age-related macular degeneration, here associated with aberrant histone modifications and alterations in chromatin structure. Indeed an age-related effect has been corroborated in the current study for miR-342 hypermethylation in CRC.

The positive correlation between miR-342 hypermethylation and the presence of chronic constipation suggests that due to the slower shedding rate of epithelial cells of the GI-track in these patients, apoptosis is prolonged, and the possibility of cancer-related changes enhanced.

One of the limitations of questionnaire-based assessments is lack of patient compliance. Moreover, inaccessibility to a homogenous statistical population from the aspects of lesion type, disease duration and type/duration of a patient’s proneness to detrimental environmental factors (such as alcohol, smoking, drug use, etc.) limits the robustness of results; increasing the sample size improving methodological accuracy when selecting a particular population is highly recommended [86,87]. In order to ensure the possibility of using miR-137 and miR-342 methylation status as a non-invasive early detection biomarker, it would be important to first repeat the study using blood samples obtained from the same lesion-bearing patients. Moreover, evaluating the methylation status of other down-regulated microRNAs in a panel may provide much more specific and sensitive results in early diagnosis of CRC epigenomics [88,89]. Ultimately, due to complex and heterogeneous pathways included in down-regulation of tumour suppressor genes in cancerous tissues, assessing the expression level of these miRNA loci in the same specimens could elucidate whether the methylation observed has caused its potential relative downregulation or not.

## 5. Conclusions

Our study strongly suggests that miR-137 and miR-342 are frequently methylated in CRC; therefore, they are a good supplement for previous studies and could be useful for better understanding of miRNA regulatory mechanisms in CRC. Results would help better appraise these putative biomarkers for detecting the lesions with higher malignancy potential, especially in the pre-symptomatic stage. This research shows an age-related effect for miR-342 hypermethylation in CRC but for a comprehensive and accurate assessment of the impact of age- and indeed sex-related factors on methylation status in CRC-related malignancies, age-matched and sex-matched controls are essential. However, research in this area is currently limited to the logistical/cultural difficulties associated with obtaining ideal sample sets.

## Figures and Tables

**Figure 1 biomolecules-09-00519-f001:**
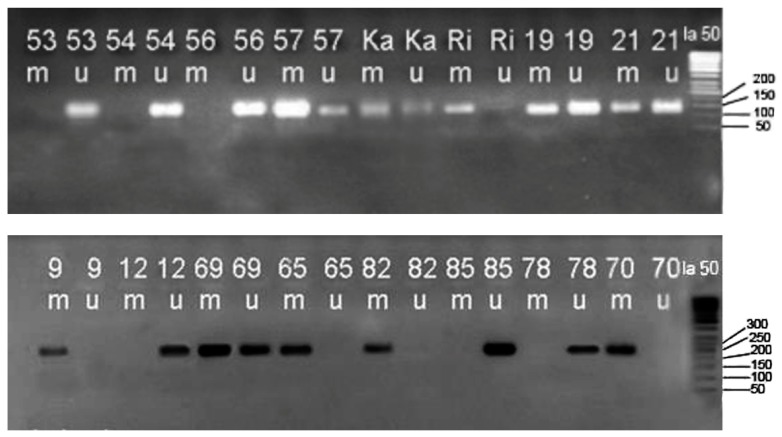
Case specimen methylation-specific polymerase chain reaction (MSP) for miR-137(A) and miR-342(B) visualized using 2% agarose gel and 50 bp ladder. For each sample, two wells were arranged named U and M wells. Presence of a band in M or U well illustrates methylation of the concordant strand. Presence of two bands in both M and U wells show Hemi methylated alleles which are believed to be due to contamination with the normal colonic mucosa.

**Figure 2 biomolecules-09-00519-f002:**
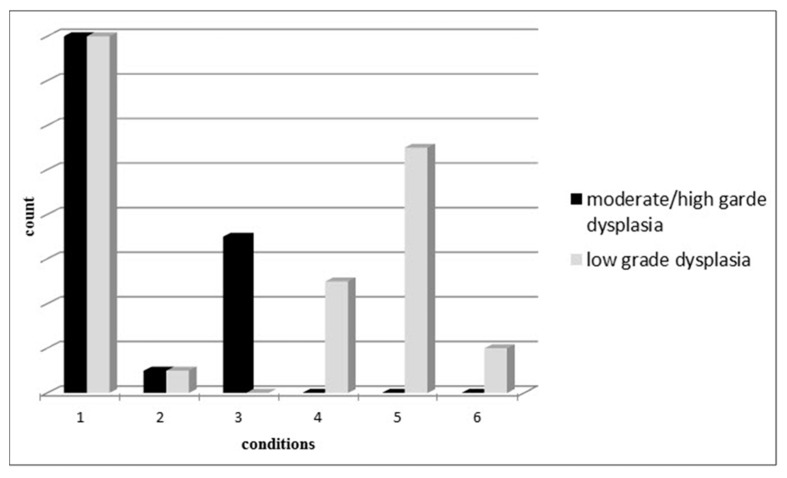
Synchronous assessment of dysplasia degree and the pathological state of the specimens.

**Table 1 biomolecules-09-00519-t001:** Reagents and volumes utilized for preparing the polymerase chain reaction (PCR) master mix.

Final Concentration	Volume/Reaction (µL)	Reagent
Distilled Water	5.45	-
10x Hot Start Buffer	1.25	1x
5X Q buffer	2.5	1x
25 mM MgCl2	0.25	1.5–5 mM
dNTP * (10 mM of each)	0.25	200 μM of each dNTP
Primer Forward (M and U)	0.75	0.1–0.5 μM
Primer Reverse (M and U)	0.75	0.1–0.5 μM
Template	1.2	1 μg/100 μl reaction
Hot Start Taq Polymerase	0.1	2.5 units/reaction
**Final Volume**	**12.5**	-

* deoxyribonucleotide triphosphate.

**Table 2 biomolecules-09-00519-t002:** Thermocycler-optimized PCR program for miR-137 and miR-342 genes.

Steps		Duration (sec.) for miR-137	Temperature (°C) for miR-342	Duration (sec.) for miR-342	Temperature (°C) for miR-342
Primary denaturation		15	95	15′	95
Cycles	Denaturation	1	95 *	30	92 **
	Annealing	35	56 *	45	64 **
	Extension	1	72 *	30	72 **
Final Extension		10	72	10	72

* Number of cycles = 40, ** Number of cycles = 45.

**Table 3 biomolecules-09-00519-t003:** Methylation-specific PCR primer properties.

Annealing Temperature (°C)	%CG	Length	Sequence	Primer
53.2	57.9	19	5′-TAGCGGTAGTAGCGGTAGC-3′	miRNA-137-M*
				(Forward)
52.4	47.6	21	5′-CTAATACTCTCCTCGACTACG-3	miRNA-137-M* (Reverse)
52.4	47.6	21	5′-TAGTGGTAGTAGTGGTAGTGG-3′	miRNA-137-U*
				(Forward)
51.1	40.9	22	5′-CTAATACTCTCCTCAACTACAC-3′	miRNA-137-U* (Reverse)
71.15–80.47	44	43	5′-GCGGTCCCAAAAGGGTCAGTTATTTTCGTTCGTTTCGTTTTTC-3′	miRNA-342-M* (Forward)
72.50–81.79	51	41	5′-GCGGTCCCAAAAGGGTCAGTAAATACGCGCGTTACTATTCG-3′	MiRNA-342-M* (Reverse)
59.51 58.99-	35	45	5′-GCGGTCCCAAAAGGGTCAGTTATTTTTGTTTGTTTTGTTTTTTGT-3′	miRNA-342-U*
				(Forward)
69.92–79.44	42	45	5′-GCGGTCCCAAAAGGGTCAGTTAAAATACACACATTACTATTCACC-3′	miRNA-342-U*
				(Reverse)

***** M: Methylated; U: Un-methylated; Underlined sequence: Universal.

**Table 4 biomolecules-09-00519-t004:** Frequency of the clinic-pathological features of the specimens.

Frequency (% and Number Out of 59)	Conditions	Clinical-Pathological Feature
Pathology report	Tubular adenomas	54.2%—(32)
	Hyperplastic polyps	25.%—(15)
	Invasive adenocarcinoma	11.9%—(7)
	Villous, juvenile and ulcerative lesions	5.1%—(3)
	Other serrated types	3.4%—(3)
Dysplasia degree	Low-grade dysplasia	59.3%—(3–5)
	High and moderate dysplasia	40.7%—(24)
Anatomic site and location in colon	Rectum and sigmoid	40.7%—(24)
	Descending colon	22%—(13)
	Transverse colon and hepatic flexure	16.9%—(10)
	Ascending colon	11.9%—(7)
	Secom	8.5%—(5)
Sex	Male	49.2% (29)
	Female	50.8% (30)
Age	17–45 yrs.	25.4% (15)
	45–65 yrs.	45.8% (27)
	More than 65 yrs.	28.8% (17)
Body mass index (BMI)	Normal (18.5–25)	27.1% (16)
	Overweight (>25)	71.2% (42)
	Underweight (<25)	1.7% (1)
Family history	Positive	28.8% (17)
	Negative	71.2% (42)
Diabetes	Positive	10.2% (6)
	Negative	89.8% (53)
Hypertension	Positive	16.9% (10)
	Negative	83.1% (49)
Nonsteroidal anti-inflammatory drugs	Positive	20.3% (12)
(NSAIDs) intake	Negative	79.7% (47)
Smoking	Positive	23.7% (14)
	Negative	76.3% (45)
Alcohol consumption	Positive	15.3% (9)
	Negative	84.7% (50)
Addiction	Positive	3.4% (2)
	Negative	96.6% (57)
Regular exercise	Positive	54.2% (32)
	Negative	45.8% (27)
Fruit-vegetable intake	Considerate	50.8% (30)
	Inappropriate	49.2% (29)
Red meat intake	Considerate	37.3% (22)
	Inappropriate	62.7% (37)

**Table 5 biomolecules-09-00519-t005:** Promoter methylation status of distinct colon lesions studied.

Pathology Report	miR-137 Methylation Status		miR-342 Methylation Status	
	Methylation Frequencies	Unmethylation Frequencies	Methylation Frequencies	Unmethylation Frequencies
Tubular adenoma polyps	15.6%	84.4%	81.3%	18.8%
Villous adenoma polyps	100%	0%	50%	50%
Invasive adenocarcinomas polyp/masses	28.6%	71.4%	75.7%	14.3%
Ulcerated, juvenile and retention polyps	20%	80%	20%	80%
Hyperplastic polyps	9.1%	90.9%	81.8%	18.2%
Other serrated polyps	0%	100%	100%	0%

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
