# Peer review of "The Differential DNA Hypermethylation Patterns of microRNA-137 and microRNA-342 Locus in Early Colorectal Lesions and Tumours"

_biomolecules, 2019, doi:10.3390/biom9100519_

Round 1

Reviewer 1 Report

it is an interesting and well designed study. However will be important to add at the introduction the role of general gene methylation on CRC (ie Anticancer Res. 2017 Mar;37(3):1105-1112; World J Gastrointest Oncol. 2017 Apr 15;9(4):142-152.)

the authors also should state clearly why they choose those miRNAs? Does these miRNAs implicated through their targets to CRC? Which are their possible targets and how could be act at the CRC carcinogenesis?

Also validation experiments are missing. It will be interesting the authors to perform ie a real time PCR experiments to confirm if the methylation status influence the miR expression

Reviewer 2 Report

Authors have performed an interesting study in regards to the presence of hypermethylated miRNA137 and 342 on colorectal cancer samples. The whole manuscript is well-written, clear and of great interest. However, I consider that Conclusions section should be improved in order to mention their findings regarding to sex and age-related differences on the methylation patterns since these data are of great interest. 

Round 2

Reviewer 1 Report

The authors cover the revision issues